# Genital Prolapse Surgery: What Options Do We Have in the Age of Mesh Issues?

**DOI:** 10.3390/jcm10020267

**Published:** 2021-01-13

**Authors:** Guenter K. Noé

**Affiliations:** 1Department of Obstetrics and Gynecology, University of Witten-Herdecke, 41540 Dormagen, Germany; g.noe2013@gmail.com or guenter.noe@uni-wh.de or karl-guenter.noe@rheinlandklinikum.de; 2Rheinlandclinics Dormagen, 41540 Dormagen, Germany

**Keywords:** pelvic floor repair, laparoscopic repair, vaginal repair, mesh use, native tissue

## Abstract

Here, we describe the current laparoscopic procedures for prolapse surgery and report data based on the application of these procedures. We also evaluate current approaches in vaginal prolapse surgery. Debates concerning the use of meshes have seriously affected vaginal surgery and threaten to influence reconstructive laparoscopic surgery as well. We describe the option of using autologous tissue in combination with the laparoscopic approach. Study data and problematic issues concerning the existing techniques are highlighted, and future options addressed.

## 1. Introduction

Vaginal prolapse is and remains a problem for a large number of women around the world. After a prolonged period of slow development, prolapse surgery progressed significantly towards the end of the 1990s. For a long time, native tissue repair dominated the surgical procedure. These techniques are advantageous as they do not require any foreign material to be implanted. A number of techniques were introduced, but were not investigated in multicenter studies. The influence of alloplastic material in the lower pelvis, its side effects, and complications were clearly underestimated. In the last two decades, mesh materials have been used to an increasing extent in pelvic floor surgery, and have almost triggered a crisis. In the last few years, we have learned more about material characteristics and behavior in tissue. As patients with weakened connective tissue cannot permanently be reconstructed, meshes are very helpful. Surgeons’ lack of experience, too little training, too little knowledge about the material properties, and incorrect indication were certainly very decisive for the mesh problem.

After the withdrawal of numerous mesh products from the market and negative media campaigns in many countries, the use of synthetic fabrics declined significantly [1]. This had a major impact on vaginal reconstructive surgery. Traditional methods such as colporraphy, the Manchester-Fothergill procedure, and sacrospinous fixation re-emerged as important approaches. Various mesh products were developed for the latter procedure, and were marketed along with clever fixation techniques. Once mesh surgery began its triumphant advance at the end of the 1990s, we were confronted with the problem of meager study data on traditional procedures that meet the current requirements.

In some countries, the use of meshes in sacropexy is viewed critically by government agencies. The technique is still recognized as the “gold standard” in prolapse surgery [2,3,4], but the extensive use of deep mesh placement is associated with greater mesh exposure and shrinkage [5]. Degradation of the material is followed by its spread within the body. Current study data indicate that these materials cause local effects on muscle, as well as fatigue syndrome [6,7]. The mesh problem in vaginal surgery has encouraged the use of native tissue and laparoscopic procedures. In the last decade, lateral suspension [8] and pectopexy [9] were introduced as alternatives to sacrocolpopexy for laparoscopic pelvic floor repair. Apart from laparoscopy, traditional vaginal procedures based on mesh materials or sewing applicators were developed further.

## 2. Vaginal Pelvic Floor Surgery

Vaginal surgery is more ambiguous than laparoscopic surgery. Regulations and practices concerning the use of meshes differ from one country to the other. All of the existing methods of vaginal mesh application are used in Germany. The indications for their use have become more stringent, but meshes are not subject to any official restrictions, although governed by a variety of regulations. The use of meshes is entirely prohibited in some countries. In others, their use is permitted in clinical studies or at selected centers. In yet other countries, it is common practice to use self-tailored meshes.

Although a number of single-center studies revealed the superiority of mesh surgery, the so-called PROSPECT (PROlapse Surgery: Pragmatic Evaluation and randomised Controlled Trials) trial showed no benefits [10]. The above-mentioned study data have been interpreted diversely. An objective presentation is almost impossible at the present time. However, research efforts are still being focused on enhancing the safety of meshes in vaginal surgery. One of the many debated issues is whether the quantity of mesh or the combination of different alloplastic tissues (such as the combined use of incontinence tapes and prolapse meshes) is a crucial determinant of success [11,12]. The second debated issue is geographical centralization of pelvic floor surgery in order to achieve better outcomes. Many studies suggest strong evidence of a close connection between surgical expertise and complication rates [13]. A large number of publications in the last decade have addressed this issue in nearly every field of surgery. The vast majority of investigations identified a clear link between success rates, complications, the number of performed interventions, and the surgeon’s expertise.

The use of meshes is nearly impossible or very limited in some countries. Traditional methods are experiencing a renaissance in these regions. For several decades, the sacrospinous ligament was used for apical fixation. Vaginal mesh surgery developed in the 1990s and current alternative methods are also based on fixation to this ligament. A Cocrane analysis in 2013 examined randomized trials that compared vaginal (especially sacrospinous fixation) and sacrocolpopexy (SC). The review disclosed the superiority of SC, but also highlighted the significantly longer operating times and the longer learning curve for SC [14]. Mesh surgery has replaced sacrospinous fixation to a significant extent. The technology of mesh fixation has been revived. It is either performed using the traditional method, or with the aid of suture devices for fixing the threads [15]. To improve the outcome, surgeons use narrow meshes instead of sutures. The meshes are fixed with sutures or anchors. Analogous to suture techniques, the anchors are placed in the ligament, close to the pudendal nerve, and are sutured at this site. All of these techniques are not performed under direct vision, and therefore require great skill. Anchors placed close to a nerve or at the wrong site may cause severe pain (Figure 1).

A rather limited body of data derived from single-center studies is available at the present time [16,17]. These studies report excellent outcomes in combination with traditional colporrhaphy. However, the majority of the published reports are short-term evaluations and do not provide data on mesh-related complications due to fibrosis or mechanical stress. The same is true of traditional methods such as the Manchester-Fothergill technique or high utero-sacral fixation. Our literature search yielded a handful of small studies and case reports. Randomized or prospective studies do not exist. The procedure of culdoplasty, frequently associated with the name McCall, is used to prevent prolapse after hysterectomy. Schiavi et al. compared two suturing techniques for culdoplasty, and noted the preventive value of both techniques. A suspension suture was performed in all patients. The study did not include a control group and provided no analysis of the general risks of a pelvic floor defect. Nevertheless, the authors conclude that the method is effective [18]. We lack any real evidence of the effectiveness of these procedures. To circumvent the use of meshes, therapy approaches based on preventive suspension have been presented at several medical conventions.

Data concerning the use of meshes in vaginal surgery are very diverse. Mesh techniques should be investigated in prospective multicenter studies in order to separate the wheat from the chaff, and to provide surgeons and patients with reliable, unequivocal data. A number of skilled surgeons use a large portfolio of techniques and are able to treat their patients effectively with low complication rates. Meshes have been used very effectively in vaginal surgery. As prolapse is a very common problem, we need effective and resilient surgical techniques with a low risk profile.

## 3. Laparoscopic Sacral Colpopexy (LSC)

This approach was first published in 1920. However, it was not until the 1960s that artificial tissue was used to bridge the distance to the sacrum. The technique is performed by placing a Y-shaped mesh posteriorly and anteriorly to the vaginal wall. The conjunction is sutured to the apical structure (cervix and vault), and the distal part of the mesh is anchored to the promontory or sacrum. It has been common practice to place the mesh as low as possible [19]. This approach is justified in terms of correct anatomical location.

The need for deep mesh placement is a debated issue. Many investigations on laparoscopic sacropexy have been performed with deep mesh implantation [20]. We lack sufficient data to provide a conclusive answer to the debated issues highlighted above. A few studies have reported on the combined use of SCP and the vaginal approach. Kaser et al. described the advantages of LCP combined with vaginal posterior colporrhaphy. In a follow-up investigation of 258 patients, the laparoscopic procedure was used in 196 patients and open laparotomy in 62 [21]. Banerjee et al. published a cohort study of patients who were treated with vaginal native tissue repair (anterior and posterior colporrhaphy), laparoscopic lateral repair, and LCP (*n* = 246). The mesh was placed between the apex and the longitudinal ligament, at the level of the first sacral vertebra [22]. After a mean period of 28 months, the re-intervention rate was 7.8%:4.8% due to de novo stress urinary incontinence (SUI) and 3% due to pelvic floor defects.

Bojahr et al. published a retrospective analysis of 301 patients treated with sacral colpopexy. In 96% of the cases, LSC was performed exclusively without deep mesh fixation or additional surgery. Approximately 4% of patients underwent vaginal colporrhaphy. Recurrent symptoms were noted in 24.7% of patients at 24.5 months after surgery [23]. This may indicate that apical fixation alone is not effective. Mesh exposure in LSC (1–5%) has been reported in many publications (1–5%) [5,24]. Computer-based models showed that straight fixation causes extensive shear forces on the pelvic fascia in LSC, and may be inferior to bilateral fixation. Further investigations on fixation techniques and mesh material will be needed to reduce the use of meshes in LSC. Other risk factors such as osteomyelitis of the promontory and frequent defecation disorders have led to the development of new strategies, as mentioned above [25,26].

Indication: In most cases, LSC is used to correct all existing defects simultaneously. In cases of cystocele, the thinned tissue of the vagina is pulled cranially. The resulting shear forces may cause tissue defects. Compensating a cystocele by pulling it cranially may result in organ displacement. This could be one explanation for the high rates of de novo SUI after LSC.

The text continues here (Figure 2).

## 4. Lateral Suspension (LS)

LS was introduced by Dubuisson in 2002. Operating times for LSC have been reported to range from 90 to 300 min. Using tackers at the promontory and dispensing with fixation at the second sacral vertebra have simplified the technique. However, the use of tackers is associated with osteomyelitis [26]. The latter is a rare condition but may develop into a real threat for the patient. Aberrant vessels or scars may hinder access to the sacrum or promontory.

The difficulties of tackers and sutures caused Dubuisson to use long mesh tapes similar to TVT (tension-free vaginal tape). A trocar is placed bilaterally at the level of the umbilicus (current modification), and graspers are introduced behind the peritoneum. The latter is undermined with the grasper in the direction of the apex. The mesh itself is fixed with absorbable tackers to the vagina and the apex. The prefabricated mesh consists of two cranial arms approximately 20 cm long and 1.5 cm wide, which are moved out extraperitoneally. The length of the arms is supposed to provide enough resistance to hold the apex without the use of sutures or tackers at the promontory [27].

Indication: (All in one repair for apical and combined prolapse). The technique is used in a similar manner as LSC, with anterior and posterior exposure. Two options are available: either the mesh is placed on the anterior vaginal wall alone, or is used by the surgeon to cover the vagina anteriorly and posteriorly. A handful of single-center studies have been published on the technique, but a randomized or multicenter trial is lacking. The large body of data from the developing center are based on hospital records and telephone interviews [28]. Patients who reported for follow-up investigations were examined physically. One year after surgery, 21.6% of patients complained of persistent prolapse symptoms. De novo incontinence was noted in 5.2%, and mesh-related complications occurred in 4.2%. The authors reported a reintervention rate of 7.3% due to symptoms of pelvic organ prolapse (POP). The long-term follow-up by telephone encompassed 51.3% of the patients, of whom 87.8% reported improvement of their symptoms after surgery.

The results of the studies reveal that the technology is producing satisfactory results. In view of the fact that the existing data are derived solely from individual centers, a multicenter study or a randomized trial would be welcome. LS is based on extensive mesh use (Figure 3). Yet, we lack data about potential long-term complications of the fibrosed arms, which cross vessels and nerves. This is a clear disadvantage in view of the perennial discussion on the use of meshes in pelvic floor surgery. The main advantage is the simplicity of the technique, which enables even less experienced laparoscopists to perform a POP correction.

## 5. Laparoscopic Pectopexy (LP)

Scientists who reported on LP in 2010 started to develop the technique in 2007 [9]. They had extensive experience in the use of LSC using the suture technique and placing the mesh on SV2 (Sacral vertebra 2). In obese patients, the distance between the mesh and the sigmoid colon is usually small because of fatty tissue. Moreover, patients frequently have a history of diverticulosis or diverticulitis. Defecation disorder rates ranging from 7% to 20% have been reported in connection with LSC [26,29]. These problems led to the development of the bilateral suspension technique using the pectineal (Cooper’s) ligament (Figure 4). The procedure was performed earlier in India through laparotomy, using the medial portion of the ligament [30]. To avoid lifting the apex towards the abdominal wall, the developer used the most cranial part of the ligament. A 15-cm PVDF (Polyvinylidenfluorid) tape was used to fix the apex bilaterally to the pectineal ligament using a suturing technique.

Indication: The technique was introduced as an apical suspension procedure with accompanying surgery to treat level-2 and level-3 defects. The so-called defect-oriented strategy enabled the surgeon to get by with little use of meshes [31].

After a first pilot study, a randomized investigation was conducted to determine potential new risks and outcomes compared with LSC. The surgical data revealed no significant differences between the two techniques, although operating times were clearly shorter for the pectopexy approach [32]. After a mean follow-up period of 21 months, a significant difference was noted with regard to defecation disorders. Moreover, significantly fewer de novo lateral defects were observed in the pectopexy arm. The overall success rate for apical support was 97.5%, and the recommendation rate 95% [33]. After the randomized trial, an international multicenter study was conducted at eleven centers in four European countries. The purpose of the study was to evaluate the safety of the technique. The data revealed a low risk for patients, and equivalent operating times as that for LSC [34]. The follow-up data also showed very satisfactory results, especially with regard to the stringent use of mesh material (see further original research).

## 6. Uterus Preservation

Indications for hysterectomy as part of prolapse surgery have changed frequently, as noted in a German study spanning the period from 1960 to 1985. Only 24.3% of the interventions were combined with hysterectomy between 1960 and 1963, while 97.7% of the interventions were combined with hysterectomy between 1978 and 1985 [35]. The overall risk profile of the intervention changed. The indications were mainly for the prevention of cancer and birth control. Hysterectomy offered no advantages in terms of long-term success. On the contrary, De Lancy emphasized the integrity of paracervical structures for the prevention of cystocele and rectocele in as early as 1992 [36]. The disadvantages of uterine preservation have not been reported so far.

In 2013, Kerbly et al. investigated reasons for hysterectomy as reported by patients; 213 women were interviewed at several centers. Only 20% of women desired a hysterectomy, while 36% were clearly opposed to it. A fifth of them would have accepted a poorer outcome, while 44% were unable to commit themselves [37]. Current reasons to preserve the uterus include the desire to have children and the desire to preserve the physical integrity of the body. Both vaginal and laparoscopic techniques are available today. While the study data for vaginal techniques (especially vaginal meshes) are still limited, sacropexy is an established procedure.

In 2013, the data of 507 women were examined retrospectively over 10 years [38]. Notable features of the study were a low complication rate of 1.8% and no mesh exposure. The hysteropexy could not be completed in 17 patients (3.4%) and 93.8% of the patients said their prolapse was “very much” or “much” better—of these women, 2.8% required repeated apical surgery.

## 7. Native Tissue Repair

Native-tissue repair, especially vaginal colporrhaphy, has long been equated with a poor outcome. According to the PROSPECT trial [10], the use of native tissue was not inferior to meshes in vaginal surgery. Barber et al. were able to show that a clinical symptom-oriented assessment of success in contrast to a strictly anatomical evaluation (pelvic organ prolapse quantification (POP-Q)) yielded very good long-term success rates for native tissue reconstruction [39]. Thus, adequate apical fixation in combination with native tissue repair reduces the use of foreign materials. These data clearly confirmed the effectiveness of native tissue repair. The limited use of meshes may also reduce complications and re-intervention rates.

## 8. Laparoscopic Native Tissue Repair

Vaginal surgery is frequently combined with the laparoscopic approach. However, in view of the variety of surgical instruments used, it would be advisable to perform native tissue repair using the laparoscopic approach. As the surgeon uses laparoscopic surgery alone, only laparoscopic instruments are required. This saves time, reduces sterilization costs, and economizes on the use of disposable instruments. Lateral repair has been known for many years, but laparoscopic treatment of anterior midline defects and posterior defects was first reported in 2018 [40,41]. The access routes for the two procedures are comparable to ventral and dorsal dissection of the vagina in sacrocolopexy, especially when placing a so-called Y-shaped mesh. In the presence of a cystocele, the tissue is usually dilated and thin. When using the vaginal approach, the vaginal mucosa must be opened and detached from the fasica in order to reach the defect. When using the laparoscopic approach, the tissue is not separated but rather compressed as a whole by sutures. Puncturing the fascia five to seven times in small increments creates a densely pleated effect when knotting. This reduces expansion and causes a significant thickening of tissue. The second effect is also useful when placing a Y-mesh to create more tissue between the mucosa and the mesh material. In addition to reducing the risk of erosion, it prevents the displacement of organs due to tension.

Native tissue repair enables the surgeon to restore the natural width and length of the vagina before the apex is adjusted in an anatomical position with minimal tension. It is not necessary to compensate a cystocele or rectocele by pulling it cranially. Similar effects can be achieved by making adjustments initially through the vaginal route. As described above, the advantage of the laparoscopic approach is that the entire tissue is retained and no vaginal scars develop. The preliminary results of the techniques have been very encouraging, but need to be substantiated in larger numbers of patients [40]. With regards exposure, the enhancement of tissue prior to the application of a mesh in LSC would also reduce risks to a significant extent.

## 9. Robotic Surgery in Urogynecology

Robotic surgery has gained significant importance in the USA. It is also being used more often in Europe and will continue to gain popularity because of new and cost-effective procedures. The remote control of the surgical instruments enables the surgeon to work without getting tired. The variety of instrumentation is very helpful to get free access to the operating field. This allows the surgeon to move safely in the working area. For several reasons, it takes time for experienced laparoscopists to perceive the advantages of robotic-assisted surgery. In addition to the laborious and time consuming procedure of docking, a number surgical steps must be modified to the new setting. On the other hand, the advantages of robotic surgery are more easily experienced by low-volume surgeons or beginners. The complexity of suturing and knotting is offset by the extensive degrees of freedom in using instruments. Less experienced surgeons are able to perform complex procedures with the aid of robotic-assisted surgery.

The main obstacle to the implementation of the technology is its cost. Robotics could be very helpful, especially in complex suturing techniques such as those used for native tissue repair. The various alternatives of robotic SC are a part of the standard repertoire at many centers. Single center studies as well as comparative studies and reviews are available in the published literature [42,43]. Robotic SC is considered equivalent to laparoscopic SP in terms of clinical and anatomical results. Generally speaking, the costs and operating times are significantly higher than those for laparoscopy. However, the time factor is of secondary importance for specialized surgeons. The positive results are comparable to the laparoscopy, and the complication rates are also equivalent. Exposure rates are similar to LSC, albeit low [24]. Robotics may be able to shorten learning times and enable more surgeons to offer minimally invasive surgery in urogynecology. Endoscopic autologous tissue reconstruction might serve as a new field for robotics.

Dealing with mesh complications and the need to remove mesh materials can be challenging. Precision and working in very small steps are essential. The support provided by robotics can be very helpful to ensure a safe approach [44].

## 10. Conclusions

The parameters that are available for the indication and the assessment of the therapeutic success (age, body mass index (BMI), general health status, sexual activity, wishes of the patient, and fears of long-term effects) require a good surgical portfolio to meet the different demands or to meet necessities. A careful diagnosis and indication are the basis for a low complication rate.

The partly serious side effects of vaginal mesh surgery have created a precarious situation in urogynecology. The use of meshes is highly restricted and even prohibited in some countries. This is a major disadvantage for patients, as meshes are required at least for the fixation of the apex. Study data indicate that patients with severely weakened connective tissue will also need tissue reinforcement with mesh materials in the future. We still use techniques that were introduced a long time ago and have never been evaluated by current standards. The existing techniques have been adapted, in part, to the skills of individual surgeons. Lateral suspension is certainly the easiest technique to perform, but also yields the most unfavorable results. Furthermore, lateral suspension has not been investigated in a randomized study or on a multicenter basis.

A large body of single-center data exist for LSC, but the implementation of the technology is very heterogeneous and therefore not comparable. LSC in particular is carried out in a wide variety of ways. Deep mesh implantation both anterior and posterior, meshes in short form, prefabricated or self-tailored meshes, the use of various sutures, and variable fixing points complicate the assessments. Hysteropexy is also performed using the Oxford technique (mesh collar), sometimes as a simple cervical fixation or as an extended posterior mesh plastic. Prospective multicenter studies are completely lacking. A major problem is the frequent and extensive use of meshes, which leads to exposure problems and reinterventions. Pectopexy is comparatively new, but offers a database extending from pilot studies to randomized studies and multicenter trials. The combination of native tissue repair and laparoscopic apex fixation permits the effective treatment of pelvic floor defects with a low risk of alloplastic materials remaining in the body. More collaborative research is needed to improve its safety for women.

Risks must be minimized in the treatment of benign disease. Training and centralization are also crucial. The ongoing improvement of treatment through competent research and standardized techniques is pursued in all surgical specialties. Independent research and development will serve as a shield against statutory restrictions. Registers should be used to optimize training and ensure consistent quality control. We need a variety of options to deal with the peculiarities of patients and the skills of surgeons. A one-fits-all strategy is not desirable. Simplifications should not be achieved at the expense of quality. Urogynecology does not belong in the hands of low-volume surgeons. Importantly, clinics should consider the entire spectrum of conservative treatment options before surgery.

## Figures and Tables

**Figure 1 jcm-10-00267-f001:**
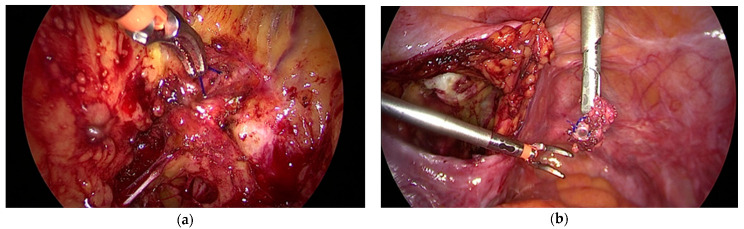
(**a**) Anchor placed in the lateral pelvic fascia instead of the sacrispinous ligament (**b**) (removed anchor on the right photograph).

**Figure 2 jcm-10-00267-f002:**
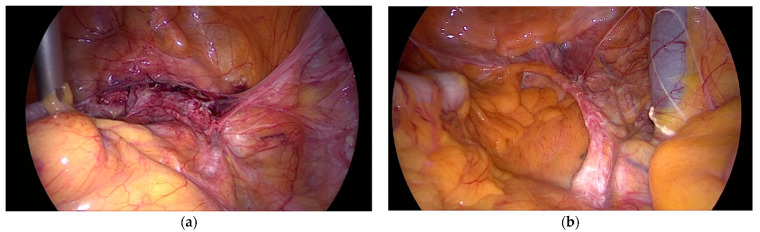
(**a**) Laparoscopic sacral colpopexy (LSC) 10 years after surgery and (**b**) 12 years after surgery, both fixed to the longitudinal ligament on SV2 (Sacral Vertebra 2).

**Figure 3 jcm-10-00267-f003:**
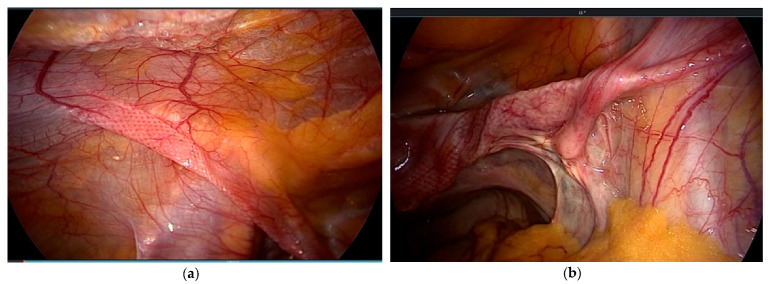
(**a**) Mesh arms extraperitoneal on the left; (**b**) crossing vessels and the psoas muscle on the right side.

**Figure 4 jcm-10-00267-f004:**
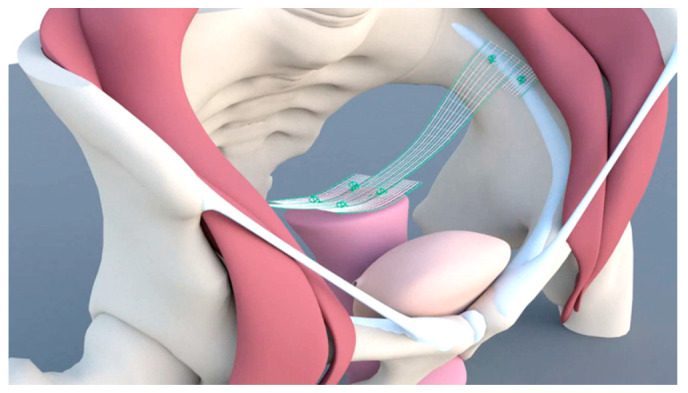
Mesh placement in pectopexy.

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
