# Peer review of "Genital Prolapse Surgery: What Options Do We Have in the Age of Mesh Issues?"

_jcm, 2021, doi:10.3390/jcm10020267_

Round 1

Reviewer 1 Report

The author performed a review of the literature and  summarised different surgical routes for  prolapse repair - especially taking into account the use of mesh . 

The topic is interesting and also important , but I have some comments 

  • Introduction : is adequate from the length but should be more structured and focus should be on different surgical routes with mesh or native tissue repair (mein advantages/disadvantages...) The reader should understand the importance and also difficulties of this topic.
  • Furthermore . I would suggest to mention indication for one or another surgical route (SKP, Lateral suspension, vag SSH...)- best way to correct posterior compartment , apex ...?
  • It is well known that pelvic floor surgery varies and no one fits all operations exist - the author should discuss or even mention that age, BMI, general health status, sexual activity, and so on also play a major role before surgeons decide which kind of pelvic floor surgery will be selected 
  • lack of evidence and heterogeneity should be discussed in more detail 
  • it should clearly be stated that this is a review of the literature - no own results or data are presented at the recent version
  • After revision and in case of acceptance this paper is suitable for the special issue " Clinical researches in urogynecology" as a review (of the literature= 

Author Response

Attached the answers to the reviewers

Reviewer 2 Report

according to our knowledge some references as  surgical management of recurrences of multicompartements pelvic organe prolapse after failure of pops :initial report of the first six cases and outcomes at 2 years follow up and robotic sacrocolpopexy by damiani et all must be cited

however llops is not well described as all the procedures and tecniques as anly short described 

another point to analize are the failure and the recurrences and how to manage it

Author Response

attached the answers to the reviewers
